# Effect of Acute and Chronic Dietary Supplementation with Green Tea Catechins on Resting Metabolic Rate, Energy Expenditure and Respiratory Quotient: A Systematic Review

**DOI:** 10.3390/nu13020644

**Published:** 2021-02-17

**Authors:** Mariangela Rondanelli, Antonella Riva, Giovanna Petrangolini, Pietro Allegrini, Simone Perna, Milena Anna Faliva, Gabriella Peroni, Maurizio Naso, Mara Nichetti, Federica Perdoni, Clara Gasparri

**Affiliations:** 1IRCCS Mondino Foundation, 27100 Pavia, Italy; mariangela.rondanelli@unipv.it; 2Department of Public Health, Experimental and Forensic Medicine, University of Pavia, 27100 Pavia, Italy; 3Research and Development Unit, Indena, 20139 Milan, Italy; antonella.riva@indena.com (A.R.); giovanna.petrangolini@indena.com (G.P.); pietro.allegrini@indena.com (P.A.); 4Department of Biology, College of Science, University of Bahrain, Sakhir Campus, Sakhir 32038, Bahrain; simoneperna@hotmail.it; 5Endocrinology and Nutrition Unit, Azienda di Servizi alla Persona “Istituto Santa Margherita”, University of Pavia, 27100 Pavia, Italy; milena.faliva@gmail.com (M.A.F.); gabriella.peroni01@universitadipavia.it (G.P.); mau.na.mn@gmail.com (M.N.); dietista.mara.nichetti@gmail.com (M.N.); federica.perdoni@unimib.it (F.P.)

**Keywords:** green tea catechins, resting metabolic rate, energy expenditure, respiratory quotient

## Abstract

The consumption of green tea catechins (GTC) is associated with modulations of fat metabolism and consequent weight loss. The aim of this systematic review was to investigate the effect of GTC on resting metabolic rate (RMR), energy expenditure (EE), and respiratory quotient (RQ). Eligible studies considered both the chronic and acute intake of GTC-based supplements, with epigallocatechin gallate (EGCG) doses ranging between 100–800 mg. Findings from 15 studies (*n* = 499 participants) lasting 8–12 weeks (for chronic consumption) or 1–3 days (for acute intake) are summarized. This review reveals the positive effects of GTC supplementation on RQ values (272 subjects). Regarding the effects of acute and chronic GTC supplementation on RMR (244 subjects) and EE (255 subjects), the results did not allow for a definitive conclusion, even though they were promising, because some reported a positive improvement (two studies revealed an increase in RMR: one demonstrated an RMR increase of 43.82 kcal/day and another demonstrated an increase of 260.8 kcal/day, mainly when subjects were also engaged in resistance training exercise). Considering GTC daily dose supplementation, studies in which modifications of energetic parameters occurred, in particular RQ reduction, considered GTC low doses (100–300 mg). GTC may be useful for improving metabolic profiles. Further investigations are needed to better define adequate doses of supplementation.

## 1. Introduction

Obesity is a common disorder with complex causes; it is manifested by a chronic energy imbalance characterized by the excessive accumulation of fat and the excessive conservation of triacylglycerol in the cells that form adipose tissue [1,2]. 

The current treatment of obesity includes reducing the calorie intake in the diet and increasing physical activity in order to increase energy expenditure (EE), but this is still not enough to reduce the trend and prevalence of obesity worldwide [3].

Recently, dietary supplements have been proposed for the management of body weight in order to counteract obesity [4].

Research in this area is expanding, with particular emphasis in the area of lipid mobilization [5]. An example of bioactive compounds present in food, which can increase the efficiency of weight loss by lipid mobilization, is green tea catechins.

Epicatechin (EC), epigallocatechin (EGC), epicatechin gallate (ECG), and epigallocatechin gallate (EGCG) are the four most abundant flavonoids in green tea; in particular, EGCG is the most active and most abundant polyphenol, as it represents about 35% of total catechins and has received attention as a potential anti-obesigenic agent [6].

In animal models, it has been demonstrated that the consumption of green tea or its main components, green tea catechins, is associated with weight loss through the modulation of fat metabolism and EE [7,8,9].

In humans (specifically obese subjects), various randomized controlled trial(RCT) have demonstrated a positive effect of green tea on weight loss or body composition [10,11,12,13,14]; green tea taken with co-supplements (capsaicin and ginger) has also been found to have beneficial effects on weight loss [15].

A 2012 Cochrane review, while not recent, claimed that green tea preparations appear to induce a small, statistically non-significant weight loss in overweight or obese adults [16], and more recent a systematic review in 2017 reported that the daily consumption of green tea with doses of EGCG between 100 and 460 mg/day shows a greater effectiveness on body fat and body weight reduction in intervention periods of 12 weeks or more [17].

These conclusions were confirmed by a more recent meta-analysis published in 2019, which suggested that the consumption of green tea supplementation in obese subjects with metabolic syndrome has beneficial effects on the improvement of lipid and glucose metabolism, as well as in the facilitation of weight loss [18].

There are several proposed mechanisms whereby green tea catechins (GTC) may influence body weight and composition. The predominant hypothesis is that GTC influence sympathetic nervous system (SNS) activity, increasing EE, and promoting the oxidation of fat. Other potential mechanisms include modifications of appetite, the up-regulation of enzymes involved in hepatic fat oxidation, and decreased nutrient absorption [19,20].

In vitro studies have shown how EGCG and ECG can be involved in the inhibition of lipogenesis, interacting with the acid-fat-synthetase enzyme [21,22,23]. A further mechanism of action studied for GTC is the inhibition of the activation of the transcription nuclear factor kappa-light-chain-enhancer of activated B cells (NF-κB), which leads to the overexpression of an enzyme involved in the β-oxidation of lipids as a final response [24,25], a mechanism that, among other things, explains the fact that the increase in fat metabolism continues over time, even after the cessation of the intake of tea catechins [26].

It was also highlighted that green tea can be involved in the processes that regulate glucose absorption [27,28,29,30].

In in vivo studies conducted on animal models, in which obesity was induced by diet or caused by a genetic origin, have shown that treatment with green tea or its polyphenolic compounds is effective in aiding the loss or maintenance of body weight [31,32,33,34,35,36].

Human studies have also shown positive results [11,20,37,38,39,40,41,42,43]. However, there has been some variability according to the population taken into consideration, such as Asian or Caucasian populations. This variability can probably be traced back to the catechol O-methyl-transferase (COMT) polymorphism [44].

The catechins in green tea may stimulate thermogenesis and fat oxidation through inhibition of COMT, an enzyme that degrades norepinephrine (NE) [45]. Studies in rats and mice have shown an EGCG-induced reduction in food intake and/or an increase in EE [46].

In humans, there has only been one meta-analysis on this topic that supports the findings that EGCG has an effect on metabolic parameters [47]. 

Given this background, this systematic review was aimed at investigating the effects of acute and chronic GTC supplementation on metabolism, in particular on resting metabolic rate (RMR), EE, and respiratory quotient (RQ).

## 2. Materials and Methods 

The present systematic review was conducted in accordance with the preferred reporting items for systematic review and meta-analyses (PRISMA) statement [48]. It was carried out through the following steps: (1) the formulation of the review question: “what are the benefits on metabolism associated with the consumption of green tea?”; (2) the defining of subjects: humans; (3) the formulation of a search strategy for the identification of relevant intervention studies that included the effect of green tea on metabolism; and (4) the analysis of the data through the systematic review.

### Search Strategy

Articles written in English were identified by searching Scopus (https://www.scopus.com/home.uri) and Google Scholar (https://scholar.google.it/). The search strategy was based on the following search terms: “green tea” AND “green tea extract” AND “resting metabolic rate” AND “basal metabolic rate” AND “energy expenditure” AND “respiratory quotient” and “substrate oxidation” AND “substrate utilization.”

The literature search was conducted in September 2020, and this search retrieved 715 studies that were entered in the flowchart process (Figure 1). Eligible studies were required to report baseline values, follow-up values, and the daily dosage of green tea used to detect effects on energy metabolism parameters, in particular RMR, EE and RQ. Eligible studies were required to consider green tea extracts with no more than 50 mg of caffeine content. Dietary supplementation was considered in both capsule and liquid forms.

## 3. Results

### 3.1. Effects of Green Tea on Resting Metabolic Rate 

Table 1 shows the effects of green tea supplement on RMR values, summarizing evidence from seven studies (244 subjects). Four of these seven studies considered the chronic intake of green tea, lasting from 8 to 12 weeks of intervention. The remaining three studies referred to an acute intake of green tea extract during the test days.

#### 3.1.1. Chronic Consumption

Regarding chronic consumption, two studies revealed an increase in RMR values after green tea extract supplementation. In particular, in the randomized controlled trial by Auvichayapat et al., at the eight week of intervention, the difference in fasting RMR was 43.82 kcal/day (183.38 kJ/day) (*p* < 0.001); the author assumed that the registered weight loss was due to the increase in EE and fat oxidation [42]. Cardoso et al. showed that the increase in RMR values of 260.8 kcal/day (1091.92 kJ/day) occurred when subjects who consumed green tea were also engaged in resistance training exercise. This was due to the significant increase in lean body mass, which is more metabolically active, and the body consequently expends more energy to maintain it. In contrast, the subjects who supplemented with green tea alone, without physical activity, showed a decrease in RMR (–270.4 kcal/day–1130.44 kJ/day) due to decreased body mass, which lowers the calorie expenditure necessary for body mass maintenance [49].

The other two studies included in the present review considering chronic assumption did not find any significant changes in fasting RMR and substrate oxidation after 8 and 12 weeks of supplementation with green tea, respectively, compared to baseline values [14,50]

#### 3.1.2. Acute Intake

Concerning acute intake, just one study showed that the tea supplement increased fasting RMR over two hours measured as the area under curve (AUC). The greatest difference between supplement and placebo was seen at one hour [51]. The other studies reported that the short-term consumption of a commercially available EGCG supplement did not increase fasting RMR [52,53].

Figure 2a,b shows that there was an overall positive correlation between posology and RMR: the coefficient was *r* = 0.797 and *r*^2^ = 0.636, The association was positive: the increase in posology was related to an increase in RMR. As shown in Figure 2a, this trend was correlated for acute and chronic intake. Between acute and chronic intake, there was not any statistically significant difference for the trend of 0.273. 

### 3.2. Effects of Green Tea on Energy Expenditure 

Table 2 shows the effects of green tea supplement on EE values, summarizing evidence from eight studies (255 subjects). Four of these eight studies considered the chronic intake of green tea lasting from 8 to 12 weeks of intervention. The remaining four studies referred to an acute intake of green tea extract during the test days.

#### 3.2.1. Chronic Intake 

Regarding chronic intake, most of the studies (three studies) included in this review revealed no statistically significant changes in EE after supplementation with green tea extract when compared to baseline values [54,55,56]. On the contrary, in the study by Chantre et al., the subjects consuming a daily dose of 375 mg of catechins (of which 270 mg were epigallocatechin gallate) showed a significant weight loss at the end of the treatment. The green tea extract stimulated thermogenesis and fat oxidation, and these results indicated the potential of GTC to influence body weight and body composition via changes in both EE and substrate utilization [57].

#### 3.2.2. Acute Intake

The same trend was observed in studies about acute intake. Only one study reported a significant increase in 24-h EE in the green tea-treated group, with a mean increase of 330 KJ; the total 24-h EE with the green tea extract was significantly higher than both the placebo and caffeine groups by 3.5% and 2.8%, respectively [26]. The other three studies reported no statistical significant changes in EE values after assuming green tea extract [58,59,60]. 

### 3.3. Effects of Green Tea on Respiratory Quotient 

Table 3 reports the effects of green tea supplement on RQ values, summarizing evidence from eight studies (202 subjects). RQ values were detected in both chronic and acute studies; three of these eight studies considered the chronic intake of green tea, and the intervention was 12 weeks. The remaining five studies referred to an acute intake of green tea extract during the test days.

#### 3.3.1. Chronic Intake 

Concerning chronic intake, only two studies considered the evaluation of RQ. One of these revealed a significant reduction in fasting RQ during supplementation with the green tea extract, suggesting an increase of fat oxidation and a decrease in carbohydrate oxidation [42,57]. This finding suggested that EGCG alone has the potential to increase fat oxidation and may thereby contribute to the anti-obesity effects of green tea. On the contrary, the study by Mielgo-Ayuso detected no significant changes in non-protein RQ and whole-body fat oxidation between the EGCG and control groups.

#### 3.3.2. Acute Intake

Of the five studies focusing on acute intake, two revealed a significant reduction of RQ values after green tea extract administration [26,60], whereas the remaining three studies did not show statistically significant differences between the supplement and placebo groups in RQ [51,53,59].

This section may be divided by subheadings. It should provide a concise and precise description of the experimental results, their interpretation, and the experimental conclusions that can be drawn.

Figure 3a,b shows that there was an overall positive correlation between the posology and RQ: the coefficient was *r* = 0.392, and *r*^2^ = 0.154, The association was positive, with an increase of posology related an increase of the RQ. As shown in Figure 3a, this trend was correlated for acute and chronic intake. Between acute and chronic intake, as shown in Figure 3b, there was not any statistically significant difference for the trend of 0.690.

### 3.4. Sensitivity Analysis

To assess the impact of study quality on effect sizes, sensitivity analyses were conducted for each study included in the present systematic review. For each scale, studies were divided into two groups (low quality and high quality) based on total quality score. When the scale developer suggested a cut-off point for low versus high quality, this was used. 

Methodological quality and sensitivity analysis showed that fifteen studies were evaluated (Table 4). Inter-rater reliability was over 3 for the Jadad scale and the quality index. The selected studies (RCT) had a median quality score of 3.67 (range 1–5) out of a maximum score of 5 on the Jadad scale. The cohort or intervention studies that used the Chalmers scale had a median quality score of 60% (range 0–10).

## 4. Discussion

The present systematic review revealed the positive effects of the supplementation of GTC on fasting and postprandial RQ values. 

As a reflection of carbohydrate and fat oxidation, RQ may be a metabolic index that predicts subsequent weight gain [63]. In general, RQ is in the range from 0.70 (complete fat oxidation) to 1.0 (complete carbohydrate oxidation), and anywhere within this range indicates that a mixture of energy substrates are simultaneously oxidized [64]. 

RQ represents a key metabolic predictor of weight gain and obesity [64]. The relationship between obesity and RQ is dynamic in response to weight changes. A high fasting RQ could be a predictor of increased body weight and fat mass over a 12-month period among young adults when compared with individuals with a low/moderate or low-RQ value [65].

Individuals with high levels of cardiorespiratory fitness and low levels of fat mass have been shown to be ‘metabolically flexible,’ so they are able to switch between glucose and fat oxidation in response to homeostatic signals, such as in postprandial or fasting condition [66].

RQ is affected by different factors, such as diet composition (high levels of carbohydrates results in elevated RQ levels), body composition (an increases in fat mass lead to an elevations of RQ) [65], and even genetic trim [67].

Given the importance of knowing the metabolic aspect of each subject, the RQ evaluation through indirect calorimetry should be the basis for personalized dietary interventions.

The observed reduction in RQ values indicates high fat oxidation and low carbohydrate oxidation, and this finding represents a promising strategy for weight loss. In a previous study by Rondanelli et al., the consumption of a combination of bioactive food ingredients (containing epigallocatechin gallate, besides capsaicins, piperine, and L-carnitine) produced a significant increase in RMR and a significant decrease in RQ, with a consequent reduction in body mass index (BMI) and fat mass values, as assessed by dual-energy X-ray absorptiometry (DXA) [68].

Regarding the effects of acute and chronic GTC supplementation on RMR and EE, the results did not allow for a definitive conclusion because some included studies reported a positive improvement but others reported no changes. Moreover, the dose–response analysis revealed that a dose-dependent association existed only for RMR, while there was no dose–effect association for RQ.

A strength of this systematic review is the fact that the included studies considered green tea extracts or, at least, very small amounts of caffeine (<50 mg of the total content of green tea extract). Recent interesting studies have investigated the effects of green tea extract supplementation on metabolic parameters and body composition, but these supplement had high caffeine contents [69,70,71]. The same observation was first carried out by Dullo et al., who revealed that the effects of the green tea extract in enhancing thermogenesis and fat oxidation could not solely be explained on the basis of its caffeine content, because treatment with an amount of caffeine equivalent to that in the extract failed to alter EE, RQ, or substrate oxidation [26]. The implication of this result is that these metabolic effects resulted from compounds other than caffeine in the green tea extract. The most likely explanation for the lack of a thermogenic effect of caffeine is that the dosage (50 mg at 3 times/day) was below the threshold for stimulating thermogenesis. Cardoso et al. considered a daily dose of 20 g of green tea with 40 mg of caffeine; the authors declared that, at these levels, neither caffeine nor theobromide presented any effect on the appraised parameters, and only the catechins presented some effects in this case. Even a meta-analysis showed that catechin–caffeine mixtures, like caffeine-only supplementation, stimulate daily energy expenditure dose-dependently when 0.4–0.5 kJ mg^−1^ was administered. However, compared with placebo, daily fat-oxidation was only significantly increased after catechin–caffeine mixture consumption [72]. This is an important finding regarding green tea extracts containing different kinds of catechins.

The present review confirmed the results obtained from a recent systematic review and meta-analysis, revealing that EGCG intake moderately reduces RQ [47]. The analyses also showed that the EGCG resulted in metabolic and EE differences, but the effects on the other measures of energy metabolism, such as fat oxidation, were relatively mild. Overall, these outcomes, together with our results, support the findings that EGCG has an effect on metabolic parameters, as previously reported by Kapoor [47].

A further consideration could be the daily dose of consumed green tea required for detecting effects on energy metabolism. The present systematic review included studies considering a wide range of EGCG daily dose supplementations, from 100 to 800 mg. Interestingly, the studies in which modifications of energetic parameters were detected, in particular RQ reduction, used low doses of EGCG, from around 100 to 300 mg. These findings were in accordance with the study of Kapoor et al., in which it emerged that EGCG alone has the potential to increase metabolic rate at a 300 mg dose [47].

This review had some limitations, such as the heterogeneity of population considered in each study and the different prescriptions allowed in parallel to the treatment (physical activities, abstention from caffeinated food, etc.). Moreover, nutritional supplements were not homogeneous, as some green tea extracts were provided as capsules and others were administered diluted in water.

Overall, the current findings support the fact that EGCG has an effect on metabolic parameters. Even though the observed effects were mild and data were limited, an improvement in energy metabolism parameters was revealed by some studies included in the current review. Thus, the intake of GTC, which contain EGCG, represents a valid plant-based and safe dietary supplementation; when administered in a proper dose, GTC supplements could be useful tools during weight loss programs because of their effects on thermogenesis stimulation and fat oxidation. The observed findings are encouraging, but further investigations are needed in order to better understand the potentiality of green tea extract on human metabolism and the adequate dose of supplementation that is useful for significative changes on metabolic parameters. 

## Figures and Tables

**Figure 1 nutrients-13-00644-f001:**
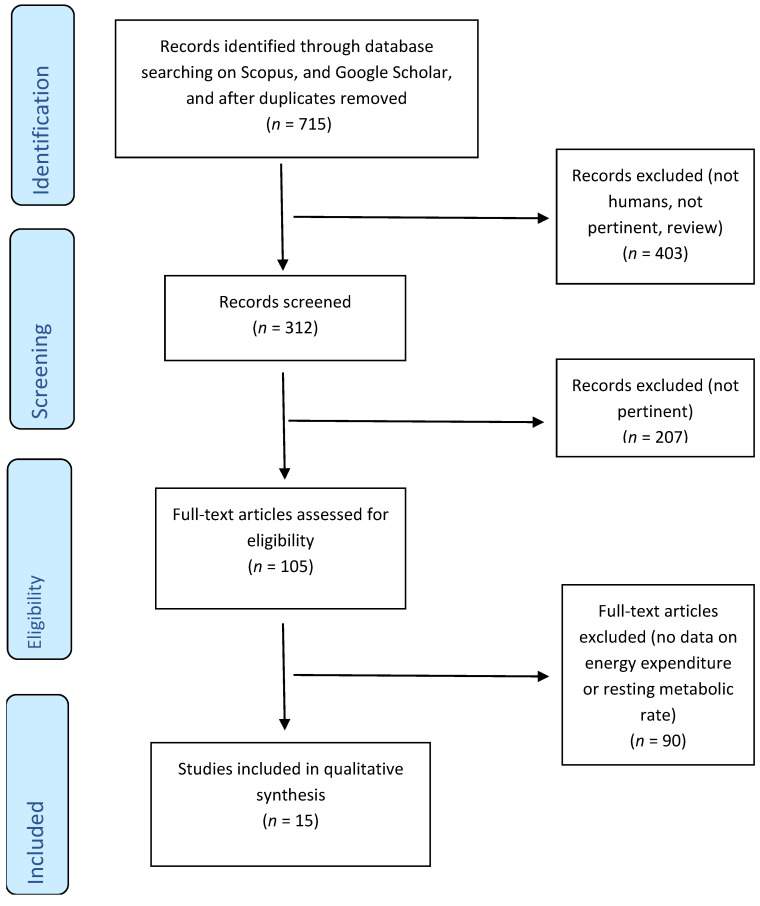
Flow diagram.

**Figure 2 nutrients-13-00644-f002:**
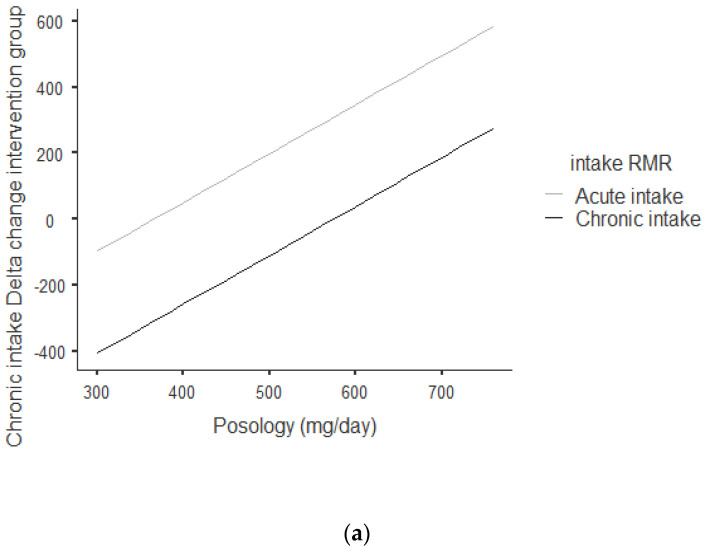
(**a**) Dose–response of acute and chronic green tea catechin intake on RMR. (**b**) Dose–response of green tea catechin intake on RMR.

**Figure 3 nutrients-13-00644-f003:**
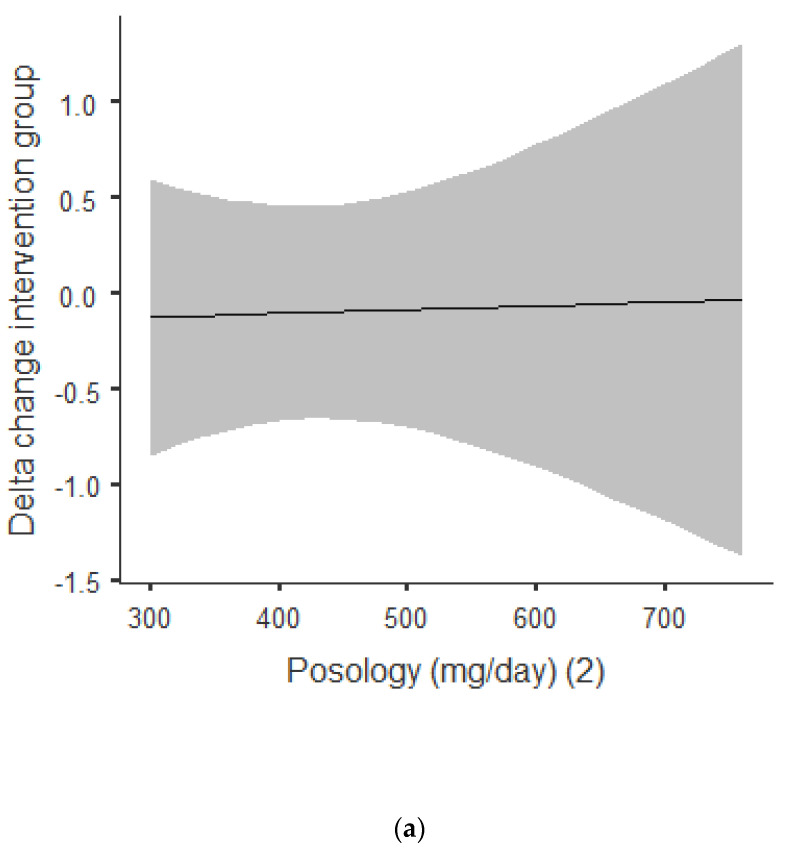
(**a**) Dose–response of green tea catechin intake on RQ. (**b**) Dose–response of green tea catechin intake on RQ.

**Table 1 nutrients-13-00644-t001:** Studies on the effects of green tea catechins (GTC) supplementation on resting metabolic rate.

First Author, Year	Study Design	Compound	Treatment and Daily Dose	Placebo or Other Treatments	Subjects	Duration	Parallel Prescriptions	Main Outcomes/Results	Method	Dietary Information
**Chronic intake**										
Quinhoneiro, 2018	Longitudinal prospective interventional Study.	Decaffeinated green tea extract capsules (Solaray®).	2 capsules daily for a total of 1009 mg of green tea leaf extract, with 450 mg of epigallocatechin gallate (EGCG), 85.7 mg of epicatechin gallate (ECG), and 5.9 mg of caffeine per day.	/	21 women with severe obesity (grade III).	8 weeks	Not to change the dietary pattern or the level of physical activity. Fast and no physical exercise or caffeine-rich beverages (coffee, black tea, or energy drink) 24 h before the assessment.	Compared to baseline values, resting metabolic rate (RMR) and substrate oxidation did not change significantly after 8 weeks of supplementation with green tea.	Indirect calorimetry QUARK-RMR device (COSMED, Rome, Italy). The QUARK-RMR device was automatically calibrated with known gas concentrations before all the assessments.	Usual diet. Macronutrient composition not specified.
Mielgo-Ayuso, 2014	Randomized, double-blind, and parallel Design.	TEAVIGO^TM^ DSM Nutritional Products.	EGCG (300 mg EGCG/day) comprising >97% of pure EGCG.	Placebo: 300 mg of lactose/day.	83 obese premenopausal women.	12 weeks.	Not to consume any other beverage containing catechins or caffeine during the intervention program. 12-week energy-restricted diet.	There were no significant interaction effects between time and treatment group (EGCG or control) on energy and substrate metabolism outcomes.	Indirect calorimetry (Vmax; SensorMedics).	55% carbohydrates, 30% lipids, and 15% proteins.
Cardoso, 2012	Double-blinded and placebo-controlled.	Sanavita Industry.	Group 1: 20 g of green tea (with 40 mg of caffeine), daily diluted in 200 ml of ice water. Group 3: green tea plus resistance training.	Group 2: 20 g of placebo. Group 4: placebo plus resistance training.	36 overweight or obese women.	8 weeks.	A base diet started 4 weeks prior to the green tea protocol was continued for all the duration of the treatment.	The RMR decreased significantly in groups 1 and 2 but increased significantly in groups 3 and 4 (*p* < 0.05). In group 1, the RMR for body mass showed no change, while in group 2, there was a measurable but not significant decline. In groups 3 and 4, the RMR for body mass increased significantly, and it was highest in group 3.	Indirect calorimetry with a pneumotachograph connected to a gas analyzer (Aerosport VO2000; Medical Graphics Corporation).	Usual diet. Macronutrient composition not specified.
Auvichayapat, 2008	Randomized controlled trial.	Herbal One®.	3 capsules daily. Each capsules contained 250 mg of green tea leaf extract, 0.24 mg of gallic acid, 4.09 mg of catechin, 28.86 mg of caffeine, 33.58 mg of EGCG, and 9.28 mg of epicatechin gallate.	Placebo: cellulose capsule.	60 obese subjects (42 females and 18 males).	12 weeks.	Thai diet (65% carbohydrates, 15% protein, and 20% fat).	Green tea increased energy expenditure by the 8th week of the study; specifically, the RMR was 372 kJ/day higher than at baseline. The increase in RMR was approximately one-third of moderate exercise. The reduction in respiratory quotient (RQ) during treatment with the green tea extract suggests that fat oxidation was higher.	The bag technique of indirect calorimetry.	Thai diet containing 65% carbohydrates, 15% protein, and 20% fat.
**Acute intake**										
Lonac, 2011	Within-subjects and repeated-measures design.	TEAVIGO^TM^ DSM Nutritional Products	135 mg of EGCG/capsule 3 times daily, for a total of 405 mg of EGCG/day. 3 capsules/day with each meal. The final capsule was consumed 2 h before metabolic testing.	135 mg of organic cornmeal/capsule, 3 times daily.	16 healthy adults (9 males and 7 females)	2 days of test. separated by a minimum of 7 days.	Test at 12-h fast, 2-h abstention from water, and 24-h abstention from exercise.	Short-term consumption did not increase RMR or the thermic effect of feeding.	Custom-built, ventilated-hood, indirect calorimetry system (Nighthawk Design, Boulder, CO).	At completion of test, liquid mixed meal (57% carbohydrate, 28% fat, and 15% protein).
Belza, 2009	Crossover, randomized, placebo-controlled, and double-blind study.	Alpine Health Products.	Group 1: tablets of 500 mg of green tea extract (where of 125 mg of catechins).	Group 2: 400 mg of tyrosine. Group 3: 50 mg of caffeine. Group 4: placebo, microcrystalline cellulose.	12 healthy, normal weight men.	4 h measurement following ingestion.	Ad libitum energy intake. Each treatment was separated by a >3-day washout.	Caffeine induced a thermogenic response of 6% above baseline value (72 ± 25 kJ per 4 h) compared to placebo (*p* < 0.0001). The thermogenic responses to GTE and tyrosine were not significantly different from placebo. Tyrosine tended to increase 4-h respiratory quotient by 1% compared to placebo (0.01 ± 0.005, *p* < 0.05).	Indirect calorimetry using a ventilated hood system.	At completion of the respiratory measurements, the subjects were given an ad libitum brunch 4 h after the intake of one of the treatment compounds. The ad libitum meal was a 1329 g pasta salad brunch (610 kJ 100 g^−1^, protein: 15 energy (E)%, carbohydrates: 55 E% and fat: 30%).
Roberts, 2005	Double-blind, placebo controlled, and crossover study.	Not specified.	The supplement contained water-soluble extracts, including 600 mg of black tea extract providing 60% polyphenols and 20% caffeine, 422 mg of guarana extract providing 36% caffeine, 100 mg of ginger extract providing 5% gingerols, 5 mg of dill weed extract, 150 mg of rutin (quercetin-3-O-glucose- rhamnose), and 50 mg of vitamin C. Each capsule contained 725 mg for a total of 1450 mg in two capsules.		16 healthy subjects (7 males and 9 females).		Subjects had no caffeine for 48 h, no exercise for 24 h, and no food for 12 h before each visit.	The area under curve (AUC) for metabolic rate increased significantly after the herbal supplement by 77.19 ± 120.10 kcal/24 h (*p* < 0.02) compared to placebo (1 kcal is equivalent to 4.184 kJ) (Figure 1). The rise in metabolic rate peaked at one hour, with no statistical significance between active and placebo arms at two hours.	Indirect calorimetry using a ventilated hood system (DeltaTrac II metabolic monitor; Datex Inc.; Helsinki, Finland).	Usual diet. Macronutrient composition not specified.

Abbreviations: RMR, resting metabolic rate; GTE, green tea extract.

**Table 2 nutrients-13-00644-t002:** Studies on the effects of GTC supplementation on total energy expenditure.

First Author, Year	Study Design	Compound	Treatment and Daily Dose	Placebo or Other Treatments	Subjects	Duration	Parallel Prescription	Main Outcomes/Results	Method	Dietary Information
**Chronic intake**										
Mahler, 2015	Randomized, double-blind, placebo-controlled, and crossover study.	TeaCare; Limmer Nutraceuticals BV.	600 mg of EGCG/day. EGCG (2 capsules × 150 mg) were taken twice daily. Metabolism was measured at rest (protocol 1) and during exercise (protocol 2).	Placebo: starch.	18 subjects with multiple sclerosis (8 males and 10 females).	12 and 4 week washout periods.	Not specified. Metabolic evaluations after a 12-h overnight fast and intake of 300 mg of EGCG in the morning.	PROTOCOL 1: Fasting energy expenditure (EE) did not differ between placebo and EGCG. PROTOCOL 2: Total increase in EE was lower in men (*p* = 0.01) and women (non-significant) receiving EGCG compared with placebo.	Canopy calorimeter (Deltatrac II; Datex Ohmeda).	50%, 35%, and 15% of energy from carbohydrates, fats, and proteins.
Brown, 2009	Double-blind, randomized, and parallel design study.	TEAVIGO™; DSM), comprising >97% pure EGCG.	800 mg of EGCG/day 2 capsules/day daily with food, one 400 mg capsule in the morning and another in the evening.	Placebo: 800 mg of lactose/day.	88 overweight or obese men.	8 weeks.	To avoid exercise and alcohol consumption for the 24 h before each measurement. Test at fast from 21.00 h on the day before. To avoid, during the intervention phase, the consumption of green tea and the dietary supplements known to affect glucose and lipid metabolism.	No significant change in total estimated energy.	not specified	Macronutrient composition not specified.
Hill, 2007	Intervention trial.	TEAVIGO.	2 capsules/day containing 150 mg of EGCG (total EGCG intake 300 mg/day)	Placebo: lactose	38 overweight or obese postmenopausal women.	12 weeks.	To maintain the normal diet and physical activity patterns (in addition to completing the required exercise sessions) during the study.	Energy intake and background energy expenditure remained constant throughout the intervention for both treatment groups.	Not specified.	Usual diet. Macronutrient composition not specified.
Chantre, 2002	Open study.	EXOLISE®.	2 times/day (i.e., 2 capsules in morning and 2 capsules at midday) a green tea extract AR25®; ingestion of 4 capsules containing AR25 provided a daily total of 375 mg of catechins, of which 270 mg were epigallocatechin gallate.		70 overweight and obese subjects (7 males and 63 females).	12 weeks.	Not specified.	Administration of green tea extract stimulated thermogenesis and fat oxidation and this extract has the potential to influence body weight and body composition via changes in both energy expenditure and substrate utilization.	Not specified.	Not specified.
**Acute Intake**										
Thielecke, 2010	Randomized, placebo controlled, and double-blind crossover trial.	TEAVIGO, a highly purified extract from green tea leaves (Camilla sinensis) containing minimum 94% EGCG and maximum 0.1% caffeine.	- Low EGCG (300 mg). - High EGCG (600 mg).	Placebo: lactose.	10 overweight or obese men.	3 days.	During the study, no caffeine and catechin-containing drinks and food were allowed.	In all the four groups, EE did not change significantly over the following 240 min before the intake of the test meal and showed an almost identical postprandial time course vs placebo.	Calorimetry, oxygen consumption (VO2, mL/min). and carbon dioxide production (VCO2, mL/min) were measured by a ventilated hood system (Deltatrac II, GE Healthcare, Freiburg, Germany). Fasting and postprandial EE.	The meal provided 5 kcal/kg body weight with 50, 35 and 15% of energy from carbohydrates, lipids and proteins, respectively.
Gregersen, 2009	Cross-over, double-blinded, and placebo-controlled design.	Unilever Colworth.	Group 1: caffeine plus a catechin mixture (600 mg) enriched in either EGCG, epigallocatechin, or a mix of catechins.	Group 2: placebo. Group 3: caffeine alone (150 mg).	15 healthy men.	1 day.	To refrain from exercise and eliminate consumption of foods or beverages containing caffeine (coffee, tea, cola, etc.) and catechins (chocolate, red wine, apples and pears) for 24 h before each chamber stay.	The multivariate ANOVA showed no significant overall treatment effect on EE.	Ventilated hood system.	Standardized meals. Macronutrient composition not specified.
Boschmann, 2007	Randomized double blind, placebo-controlled, and cross-over pilot study.	TEAVIGO™ containing at a minimum 94% EGCG and at a maximum 0.1% caffeine.	Two capsules of 150 mg of EGCG daily for 2 days prior to testing.	Not specified.	6 overweight men.	2 days.	Not specified.	After intake of the meal, EE changed in the same way with both treatments, i.e., a strong initial increase followed by a slow decrease towards the end of the measurement.	Indirect calorimetry with a canopy device.	Test meal containing 5 kcal/kg body weight with 50%, 35%, and 15% energy from carbohydrates, lipids, and proteins, respectively.
Dullo, 1999	Randomized controlled trial.	Capsules containing the green tea extract AR25 provided daily a total of 150 mg of caffeine and 375 mg of catechins, of which 270 mg were epigallocatechin gallate.	1) A green tea extract containing 50 mg of caffeine and 90 mg of epigallocatechin gallate, 2 capsules 3 times/day.	2) 50 mg of caffeine. *3*) A placebo that consisted of cellulose as inert filler.	10 healthy men.	3 separate occasions.	Not specified.	Relative to placebo, treatment with the green tea extract resulted in a significant increase in 24-h EE (4%; *p* < 0.01) and a significant decrease in 24-h RQ (from 0.88 to 0.85; *p* < 0.001).	Indirect calorimetry during the stay in the respiratory chamber.	13% of energy as protein, <40% as fat, and <47% as carbohydrates.

Abbreviations: EGCG, epigallocatechin gallate; VO2, oxygen consumption; VCO2, carbon dioxide production; ANOVA, analysis of variance; EE, energy expenditure; RQ, respiratory quotient.

**Table 3 nutrients-13-00644-t003:** Studies on the effects of GTC supplementation on respiratory quotient.

First Author, Year	Study Design	Compound	Treatment and Daily Dose	Placebo or Other Treatments	Subjects	Duration	Parallel Prescriptions	Main Outcomes/Results
**Chronic intake**								
Mielgo-Ayuso, 2014	Randomized, double-blind, and parallel design.	TEAVIGO^TM^ DSM Nutritional Products.	EGCG (300 mg of EGCG/day) comprising >97% of pure EGCG.	Placebo: 300 mg of lactose/day.	83 obese premenopausal women.	12 weeks.	Not to consume any other beverage containing catechins or caffeine during the intervention program. 12-week energy-restricted diet.	The changes in NPRQ and whole-body fat oxidation did not significantly differ between the EGCG and control groups.
Auvichayapat, 2008	Randomized, controlled trial.	Herbal One®.	3 capsules daily. Each capsule contained 250 mg of green tea leaf extract, 0.24 mg of gallic acid, 4.09 mg of catechin, 28.86 mg of caffeine, 33.58 mg of EGCG, and 9.28 mg of epicatechin gallate.	Placebo: cellulose capsule.	60 obese subjects (42 females and 18 males).	12 weeks.	Thai diet (65% carbohydrates, 15% protein, and 20% fat).	The reduction in RQ during treatment with the green tea extract suggested that fat oxidation was higher.
**Acute intake**								
Belza, 2009	Crossover, randomized, placebo-controlled, and double-blind study.	Alpine Health Products	Group 1: tablets of 500 mg of green tea extract (where 125 mg were catechins).	Group 2: 400 mg of tyrosine. Group 3: 50 mg of caffeine. Group 4: placebo, microcrystalline cellulose.	12 healthy, normal weight men	4 h measurement following ingestion	Ad libitum energy intake. Each treatment was separated by >3-day washout	There was no periodic effect on 4 h RQ or any interaction between treatments and the previous treatment (carry-over effect).
Gregersen, 2009	Cross-over double-blinded placebo-controlled design.	Unilever Colworth.	Group 1: caffeine plus a catechin mixture (600 mg) enriched in either EGCG, epigallocatechin. or a mix of catechins.	Group 2: placebo. Group 3: caffeine alone (150 mg).	15 healthy men.	1 day.	To refrain from exercise and eliminate the consumption of foods or beverages containing caffeine (coffee, tea, cola, etc.) and catechins (chocolate, red wine, apples, and pears) for 24 h before each chamber stay.	No significant treatment effect on RQ.
Boschmann, 2007	Randomized double blind, placebo-controlled, cross-over pilot study.	TEAVIGO™ Containing, at a minimum, 94% EGCG and, at a maximum, 0.1% caffeine.	Two capsules of 150 mg of EGCG daily for 2 days prior to testing.	Not specified.	6 overweight men.	2 days.	Not specified.	During the first postprandial monitoring phase, RQ values were significantly lower with EGCG compared to the placebo.
Roberts, 2005	Double-blind, placebo controlled, and crossover study.	Not specified.	The supplement contained water-soluble extracts, including 600 mg of black tea extract providing 60% polyphenols and 20% caffeine, 422 mg of guarana extract providing 36% caffeine, 100 mg of ginger extract providing 5% gingerols, 5 mg of dill weed extract, 150 mg of rutin (quercetin-3-O-glucose- rhamnose), and 50 mg of vitamin C. Each capsule contained 725 mg for a total of 1450 mg in two capsules.		16 healthy subjects (7 males and 9 females).		Subjects had no caffeine for 48 h, no exercise for 24 h, and no food for 12 h before each visit.	There were no statistically significant differences between the supplement and placebo groups in RQ.
Dullo, 1999	Randomized controlled trial.	Capsules containing the green tea extract AR25 provided daily a total of 150 mg of caffeine and 375 mg of catechins, of which 270 mg were epigallocatechin gallate.	1) a green tea extract containing 50 mg of caffeine and 90 mg of epigallocatechin gallate, 2 capsules 3 time/day.	2) 50 mg of caffeine. 3) A placebo that consisted of cellulose as inert filler.	10 healthy men.	3 separate occasions.	Not specified.	Treatment with the green tea extract yielded significantly lower values than did the other 2 treatments during all 3 periods.

Abbreviations: EGCG, epigallocatechin gallate; NPRQ, non-protein respiratory quotient; RQ, respiratory quotient.

**Table 4 nutrients-13-00644-t004:** Sensitivity analysis.

First author, Year	Quality Scale	Definition of High Quality	Quality
Quinhoneiro, 2018	Chalmers [61]	Score >50 % of total possible score	High
Mielgo-Ayuso, 2014	Jadad [62]	Score ≥3 out of 5	High
Cardoso, 2012	Jadad [62]	Score ≥3 out of 5	High
Auvichayapat, 2008	Jadad [62]	Score ≥3 out of 5	High
Lonac, 2011	Chalmers [61]	Score >50 % of total possible score	High
Belza, 2009	Jadad [62]	Score ≥3 out of 5	High
Roberts, 2005	Chalmers [61]	Score >50 % of total possible score	High
Mahler, 2015	Jadad [62]	Score ≥3 out of 5	High
Brown, 2009	Jadad [62]	Score ≥3 out of 5	High
Hill, 2007	Chalmers [61]	Score >50 % of total possible score	Low
Chantre, 2002	Chalmers [61]	Score >50 % of total possible score	Low
Thielecke, 2010	Jadad [62]	Score ≥3 out of 5	High
Gregersen, 2009	Jadad [62]	Score ≥3 out of 5	High
Boschmann, 2007	Jadad [62]	Score ≥3 out of 5	High
Dullo, 1999	Jadad [62]	Score ≥3 out of 5	High

## Data Availability

Not applicable.

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
