# Peer review of "Effect of Acute and Chronic Dietary Supplementation with Green Tea Catechins on Resting Metabolic Rate, Energy Expenditure and Respiratory Quotient: A Systematic Review"

_nutrients, 2021, doi:10.3390/nu13020644_

Round 1

Reviewer 1 Report

The review by Rondanelli et al. evaluates the effect of green tea catechins on RMR and QR. It is an excellent addition to the recent review of Li et al. on the benefit of the same molecules on various metabolic syndrome markers. The manuscript is easy to read and informative. The tables are essentials and well presented.

  • An assessment of the heterogeneity of the recording methods is mandatory. The paper presents data obtained with optic O2 sensors at the same confidence level as those obtained with paramagnetic O2 sensor. Similarly, some systems are calibrated daily with reference gases while others are not calibrated and rely only on automatic calibration features. Furthermore, some experiments use a Douglas bag, other a canopy, while several were achieved with a face mask. All these variations in the recording methods and the temporal aspects of the recordings are likely to introduce a significant bias probably more considerable than the green tea extracts themselves. We suggest that (i) the exact methods (including the calibration step) are included in the tables and (ii) that a comprehensive analysis of the biais was included in the manuscript.
  • A sensitivity analysis is also required to ascertain the effects of the primary outcomes.
  • The relationship between RQ and the obesity outcome is more complicated than what the authors described line 186. Updated references, including the ambiguous relationship between RQ and weight gain, must be inserted in the discussion.

Author Response

Manuscript entitled “Effect of acute and chronic dietary supplementation with green tea catechins on resting metabolic rate, energy expenditure and respiratory quotient: a systematic review”

To Editor-in-Chief,

We revised the manuscript with modifications and changes based on the new reviewer’s comments.

We send you the revised manuscript together with our point-by-point response.

The changes in the text are green highlighted.

Thanking you in advance for your kind collaboration and suggestions.

Best regards,

REVIEWER 1

The review by Rondanelli et al. evaluates the effect of green tea catechins on RMR and QR. It is an excellent addition to the recent review of Li et al. on the benefit of the same molecules on various metabolic syndrome markers. The manuscript is easy to read and informative. The tables are essentials and well presented.

QUESTION

An assessment of the heterogeneity of the recording methods is mandatory. The paper presents data obtained with optic O2 sensors at the same confidence level as those obtained with paramagnetic O2 sensor. Similarly, some systems are calibrated daily with reference gases while others are not calibrated and rely only on automatic calibration features. Furthermore, some experiments use a Douglas bag, other a canopy, while several were achieved with a face mask. All these variations in the recording methods and the temporal aspects of the recordings are likely to introduce a significant bias probably more considerable than the green tea extracts themselves. We suggest that (i) the exact methods (including the calibration step) are included in the tables and (ii) that a comprehensive analysis of the biais was included in the manuscript.

ANSWER

A specific column about this topic has been added in table 1 and table 2.

QUESTION

A sensitivity analysis is also required to ascertain the effects of the primary outcomes.

 ANSWER

A Sensitivity analysis has been added and it’s reported in table 4.

QUESTION

The relationship between RQ and the obesity outcome is more complicated than what the authors described line 186. Updated references, including the ambiguous relationship between RQ and weight gain, must be inserted in the discussion.

ANSWER

A paragraph about this topic has been added in the discussion section.

Reviewer 2 Report

This study summarized the effects of green tee catechins on energy expenditure measures.

The reviewer would like to highly acknowledge the attempt. However, several critical concerns should be addressed as below.

Major comments:

  1. [Abstract]line 27, “This review revealed positive effects of GTC supplementation on RQ values” Please indicate what the phase of RQ values the author was indicating. Fasting RQ? Postprandial RQ? 24hRQ?
  2. [Methods] Search words could be improved. At least the authors should use “or” like this: “green tee” and “resting metabolic rate” or “energy expenditure”. Also, search words would be too few. Let's say, please include “basal metabolic rate”, “respiratory quotient”, “specific dynamic action”, “metabolic rate”, “substrate oxidation” “substrate utilization” “fat oxidation”, “fat utilization” “oxidized fat” etc…
  3. [Methods]please indicate when did the authors search those words using the academic search engines. Otherwise, the readers and the reviewers can not judge the right of this review.
  4. [Methods][Results] Also, the reviewer has noticed that there is a lack of important and the latest studies indicating the association of objective of the present review. E.g. Janssens et al. JN 2015, Yoneshiro AJCN 2017, Katada et al. EJN 2020. Please re-study carefully.
  5. [Methods][Results] please indicate the results using the 2*3 methods: Intervention periods (Acute vs. Chronic) and measurement methods (RMR, several hours EE after ingestion, and 24hEE) .
  6. [Results] please analyze the dose-response of green tea catechin and display the results visually(graphically) for the readers.
  7. [Results][Discussion]For evaluation of fat oxidation or RQ in humans, controlling macronutrient composition at least several days before the experiment day is the most important. Please indicate and discuss the dietary information from several days before the experiment day.
  8. [Results] The authors seem to incorrectly understand? or just use? the terminology of human metabolic parameters. In table 1, the authors are indicating the effect on RMR but in table 2, the authors are also indicating fasting EE or resting EE. What are the differences among these words? Please re-organize terminology.

Minor comments:

  1. [Abstract]line 31, The reviewer believes the unit of the value should be per day (i.e. kcal/day)
  2. [Abstract]line 33, please use EE instead of energy expenditure, if you Abbreviated before. Please check up on this kind of mistakes throughout. The reviewer has noticed the same issue in the Introduction section.

Author Response

Manuscript entitled “Effect of acute and chronic dietary supplementation with green tea catechins on resting metabolic rate, energy expenditure and respiratory quotient: a systematic review”

To Editor-in-Chief,

We revised the manuscript with modifications and changes based on the new reviewer’s comments.

We send you the revised manuscript together with our point-by-point response.

The changes in the text are green highlighted.

Thanking you in advance for your kind collaboration and suggestions.

Best regards,

REVIEWER 2

This study summarized the effects of green tee catechins on energy expenditure measures.

The reviewer would like to highly acknowledge the attempt. However, several critical concerns should be addressed as below.

Major comments:

  1. QUESTION

[Abstract]line 27, “This review revealed positive effects of GTC supplementation on RQ values” Please indicate what the phase of RQ values the author was indicating. Fasting RQ? Postprandial RQ? 24hRQ?

ANSWER

The phase of RQ has been added.

  1. QUESTION

[Methods] Search words could be improved. At least the authors should use “or” like this: “green tee” and “resting metabolic rate” or “energy expenditure”. Also, search words would be too few. Let's say, please include “basal metabolic rate”, “respiratory quotient”, “specific dynamic action”, “metabolic rate”, “substrate oxidation” “substrate utilization” “fat oxidation”, “fat utilization” “oxidized fat” etc…

ANSWER

Search strategy has been improved.

  1. QUESTION

[Methods]please indicate when did the authors search those words using the academic search engines. Otherwise, the readers and the reviewers can not judge the right of this review.

ANSWER

The period of the literature search has been added.

  1. QUESTION

[Methods][Results] Also, the reviewer has noticed that there is a lack of important and the latest studies indicating the association of objective of the present review. E.g. Janssens et al. JN 2015, Yoneshiro AJCN 2017, Katada et al. EJN 2020. Please re-study carefully.

ANSWER

These studies has been cited.

  1. QUESTION

[Methods][Results] please indicate the results using the 2*3 methods: Intervention periods (Acute vs. Chronic) and measurement methods (RMR, several hours EE after ingestion, and 24hEE) .

ANSWER

A better specification of the method has been added when results are described.

  1. QUESTION

[Results] please analyze the dose-response of green tea catechin and display the results visually(graphically) for the readers.

ANSWER

A graphic about the dose response of green tea catechin has been added. The dose-response analysis revealed that a dose-dependent association exist only for RMR, while for RQ there is no dose-effect association.

Here attached, the relative table with data.

Figure 1 a and b show that there is an overall positive correlation between the posology and the RMR: the coefficient is r 0.797   and r2=0.636, The association is positive, increase the posology, increase the RMR change. As showed in figure (a) this trend is correlated for acute and chronic intake.  Between acute and chronic intake, there is not any statistically significant difference for trend 0.273. Table 1 shows the Estimated Marginal Means - Posology (mg/day) ✻ intake RMR.

Table 1. Estimated Marginal Means - Posology (mg/day) ✻ intake RMR

95% Confidence Interval

intake RMR

Posology (mg/day)

Marginal Mean

SE

Lower

Upper

Acute intake

343

-33.5

207

-691

624

501

μ

199.9

163

-319

719

658

433.3

206

-222

1088

Chronic intake

343

-342.2

206

-997

313

501

μ

-108.8

163

-628

410

658

124.6

207

-533

782

Note. ⁻ mean - 1SD, μ mean, ⁺ mean + 1SD

Figure 2 a and b show that there is an overall positive correlation between the posology and the RQ: the coefficient is r 0.392   and r2=0.154, The association is positive, increase the posology, increase the RQ change. As showed in figure (a) this trend is correlated for acute and chronic intake.  Between acute and chronic intake as showed in figure (b), there is not any statistically significant difference for trend 0.690 Table 2 shows the Estimated Marginal Means - Posology (mg/day) ✻ intake RQ.

Table 2. Estimated Marginal Means - Posology (mg/day) (2) ✻ intake rq

95% Confidence Interval

intake rq

Posology (mg/day) (2)

Marginal Mean

SE

Lower

Upper

Acute intake

256

-0.04375

0.216

-0.974

0.886

441

μ

-0.00777

0.214

-0.927

0.912

626

0.02821

0.307

-1.291

1.348

Chronic intake

256

-0.21640

0.264

-1.352

0.919

441

μ

-0.18042

0.171

-0.915

0.555

626

-0.14444

0.195

-0.984

0.696

Note. ⁻ mean - 1SD, μ mean, ⁺ mean + 1SD

  1. QUESTION

[Results][Discussion]For evaluation of fat oxidation or RQ in humans, controlling macronutrient composition at least several days before the experiment day is the most important. Please indicate and discuss the dietary information from several days before the experiment day.

ANSWER

A specific column about this topic has been added in table 1 and table 2.

  1. QUESTION

[Results] The authors seem to incorrectly understand? or just use? the terminology of human metabolic parameters. In table 1, the authors are indicating the effect on RMR but in table 2, the authors are also indicating fasting EE or resting EE. What are the differences among these words? Please re-organize terminology.

ANSWER

Terminology along the whole text has been revised. The correct abbreviations are: RMR (resting metabolic rate) and EE (energy expenditure).

Minor comments:

  1. QUESTION

[Abstract]line 31, The reviewer believes the unit of the value should be per day (i.e. kcal/day)

ANSWER

This unit along the whole text has been modified: “die” has been modified in “day”.

  1. QUESTION

[Abstract]line 33, please use EE instead of energy expenditure, if you Abbreviated before. Please check up on this kind of mistakes throughout. The reviewer has noticed the same issue in the Introduction section.

ANSWER

The abbreviations have been corrected throughout the text. We use the abbreviated term, after having dissolved the abbreviation on first use.

Reviewer 3 Report

This review manuscript summarized the effect of acute and chronic dietary supplementation with green tea catechins on resting metabolic rate energy expenditure and respiratory quotient. There are some major concerns:

  1. The abstract is a mess. Please rewrite it.
  2. The review paper is not a shopping list. It is not only a summary of the other’s research but need to extract the ideas, progress in the field.

It needs a thoroughly rewrite.

Author Response

Manuscript entitled “Effect of acute and chronic dietary supplementation with green tea catechins on resting metabolic rate, energy expenditure and respiratory quotient: a systematic review”

To Editor-in-Chief,

We revised the manuscript with modifications and changes based on the new reviewer’s comments.

We send you the revised manuscript together with our point-by-point response.

The changes in the text are green highlighted.

Thanking you in advance for your kind collaboration and suggestions.

Best regards,

REVIEWER 3

This review manuscript summarized the effect of acute and chronic dietary supplementation with green tea catechins on resting metabolic rate energy expenditure and respiratory quotient. There are some major concerns:

QUESTION

The abstract is a mess. Please rewrite it.

ANSWER

The abstract has been rewritten

QUESTION

The review paper is not a shopping list. It is not only a summary of the other’s research but need to extract the ideas, progress in the field.

It needs a thoroughly rewrite.

ANSWER 

Many parts of the text have been rewritten and a conclusion has been added to each chapter.

Round 2

Reviewer 1 Report

The paper is acceptable for publication without any additional corrections.

Author Response

Thank you for your opinion..

The authors

Reviewer 2 Report

The manuscript seems to be improved by revising. However, re-revise the below points and re-check the manuscript throughout.  

  1. Still, there is /die, not /day in the Abstract.
  2. Searching words” (((((((green tea) OR (green tea extract)) AND (resting metabolic rate)) AND (basal metabolic rate)) AND (energy expenditure)) AND (respiratory quotient)) AND (substrate oxidation)) AND (substrate utilization)” should not work on the PubMed engine. At least, when I tried to do it, it did not work. Please re-consider it or delete “PubMed”. 

Author Response

The manuscript seems to be improved by revising. However, re-revise the below points and re-check the manuscript throughout.  

  1. Still, there is /die, not /day in the Abstract.

Answer: “Die” has been changed in “day”.

  1. Searching words” (((((((green tea) OR (green tea extract)) AND (resting metabolic rate)) AND (basal metabolic rate)) AND (energy expenditure)) AND (respiratory quotient)) AND (substrate oxidation)) AND (substrate utilization)” should not work on the PubMed engine. At least, when I tried to do it, it did not work. Please re-consider it or delete “PubMed”. 

Answer: Pubmed has been deleted and search terms has been improved and made easier.

Reviewer 3 Report

This revision improved a lot. Some comments bellow:

1) The authors should clearly bring out their perspectives in this research field.

2) At line 108, so many brackets. Please clarify or make a easy way.

Author Response

Pavia, February 04, 2020

Manuscript entitled “Effect of acute and chronic dietary supplementation with green tea catechins on resting metabolic rate, energy expenditure and respiratory quotient: a systematic review”

To Editor-in-Chief,

In this round of review, we revised the manuscript with modifications and changes based on the Reviewers’ comments.

We send you the revised manuscript together with our point-by-point response.

This round changes in the text are highlighted in yellow.

Thanking you in advance for your kind collaboration and suggestions.

Best regards,

The authors

Reviewer 3

This revision improved a lot. Some comments bellow:

1) The authors should clearly bring out their perspectives in this research field.

Answer: Personal perspectives have been added in the discussion section.

2) At line 108, so many brackets. Please clarify or make a easy way.

Answer: Search terms has been improved and made easier.